# Outcomes of fenestrated endovascular abdominal aortic repair in distal entry tears of chronic debakey IIIb aortic dissection

Chi Cui<sup>☯</sup>, Bisi Wang<sup>☯</sup>, Wei Liu *

Center of Vascular and Interventional Surgery, Department of General Surgery, The Third People's Hospital of Chengdu, Affiliated Hospital of Southwest Jiaotong University & The Second Affiliated Hospital of Chengdu, Chongqing Medical University, Chengdu, China

☯ These authors contributed equally to this work.
* 1033825521@qq.com

**Data Availability Statement:** All relevant data are within the manuscript and its Supporting information files.

**Funding:** The author(s) received no specific funding for this work.

## Abstract

Currently, there have been very few reports within the literature which specifically address using fenestrated and branched stent grafts to completely isolate and repair distal entry tears of chronic DeBakey IIIb aortic dissection. This study aimed to evaluate the clinical outcomes of a 3-dimensional (3D) printed aortic model-guided fenestrated stent in the treatment of distal tears of chronic DeBakey IIIb aortic dissection after thoracic endovascular aortic repair (TEVAR). The study was a one-center retrospective study comprising 36 patients who underwent TEVAR and fenestrated endovascular abdominal aortic repair (F-EVAR) between April 2014 and December 2022. Patient data was compiled and analysed for preoperative, intraoperative, and perioperative characteristics. In total, 36 patients (12 females and 24 males) were incorporated into this study. All of the patients included in this study had hypertension, and among them, the leading cause for undergoing II-stage F-EVAR was the progression of a false lumen, accounting for 24 cases (66.7% of the total). The technical success rate was 97.2% and there were no cases of 30-day mortality, myocardial infarction, permanent paraparesis, or organ failure. One year post-F-EVAR treatment, surviving patients showed significant false and true lumen remodelling with 100% complete false-lumen thrombosis. A total of five patients died during the follow-up, two patients died related to aorta complications and three patients died of heart failure, multiple organ failure, or septic shock. II-stage F-EVAR was safe and feasible operation to repair all distal tears of chronic DeBakey IIIb aortic dissection.

## Introduction

Thoracic endovascular aortic repair (TEVAR) is the preferred treatment for type B aortic dissection (TBAD) [1, 2]. However, standard TEVAR only repairs proximal tears, leaving distal tears and residual dissection to heal without surgical assistance [3]. Previous studies have revealed that aortic remodelling after TEVAR predominantly occurs at the level of the thoracic aorta with false lumen thrombosis (FLT) occurring in only 23% of patients with dissection

**Competing interests:** The authors have declared that no competing interests exist.

involving the abdominal aorta [4, 5]. Therefore, residual aortic dissection after TEVAR, including the abdominal aorta and the iliac artery segments remains a concern. Using fenestrated and branched stent grafts to completely isolate and repair aortic tears has been determined to be the most effective treatment for residual aortic dissection. However, this treatment has rarely been reported in the literature. Therefore, the purpose of this study was to evaluate the clinical outcomes of a 3-dimensional (3D) printed aortic model-guided fenestrated stent in the treatment of distal tears after TEVAR.

## Methods

### Patients

The clinical data of all TBAD patients who underwent TEVAR at our centre from April 2014 to December 2022 were reviewed. All data were collected in February 2023 and were fully anonymized before we accessed them. Criteria for patients who were to undergo fenestrated endovascular abdominal aortic repair (F-EVAR) included those experiencing complications from residual aortic dissection such as organ ischemia, false lumen progression (>5 mm in 6 months), new stent-induced entry tears, and post-dissection aneurysm (> 5cm) during their follow-up. Patients with incomplete data were excluded. All included patients were considered unsuitable for open surgical intervention. A total of 36 patients were enrolled in the study. Patients signed the informed consent that their clinic data will be used in medical study before admitted to hospital. The study was approved by the ethics committee of The Third People's Hospital of Chengdu.

### Procedures

A 3D-printed model was constructed according to the aortic computed tomography angiography (CTA) of each patient. Fabrication of the 3D printed model: (1) Computed tomography data were imported into EndoSize software for 3D reconstruction. (2) Reverse engineering computer-assisted design software was used to reconstruct the nonparametric surface of the vessels to obtain a mathematical model of the vessels. (3) After the simulation analysis, the positions of the fenestrations for the main arterial branches were determined. (4) The 3D printed guide plate was designed and sent to the 3D printer for manufacturing of the model. (5) The printed 3D model was washed and cured to achieve transparency and stability.

The main aortic covered stent (Ankura, Shenzhen, China) was completely released in the 3D printed model on the sterile platform and the position of each fenestration was marked on the stent according to the model (Fig 1A). Four fenestrations were created with an electric pen (CIRX, Inc, Ningbo, China) and were reinforced with a platinum spring coil (Cook Medical, Bloomington, IN, USA) at the edge of each fenestration. The graft was then reattached to the delivery system and partially restrained using a V18 guide wire. After general anesthesia, patient had normally the aortic stent delivered through the right femoral artery and gradually released in the abdominal aorta to concentrate on the aortic visceral branches. Celiac trunk (Fig 1B), superior mesenteric artery (Fig 1C) and bilateral renal artery (Fig 1D) were superselected through bilateral brachial artery. The covered stent (Gore Viabahn) was deployed through the fenestration into the arteriorenal trunk and four fifths into the vessel. The balloon dilation stent (Invatec Hippocampus) was then inserted to be placed at the distal end of the arteriorenal trunk. Either a bare stent (Covidien Ever Flex) or a covered stent (Gore Viabahn) was implanted into the superior mesenteric artery and the celiac trunk. The whole segment of the fenestrated aortic-covered stent was released, and the bundle diameter guide wire was removed. A covered stent was used to isolate and repair all the tears in the aorta.

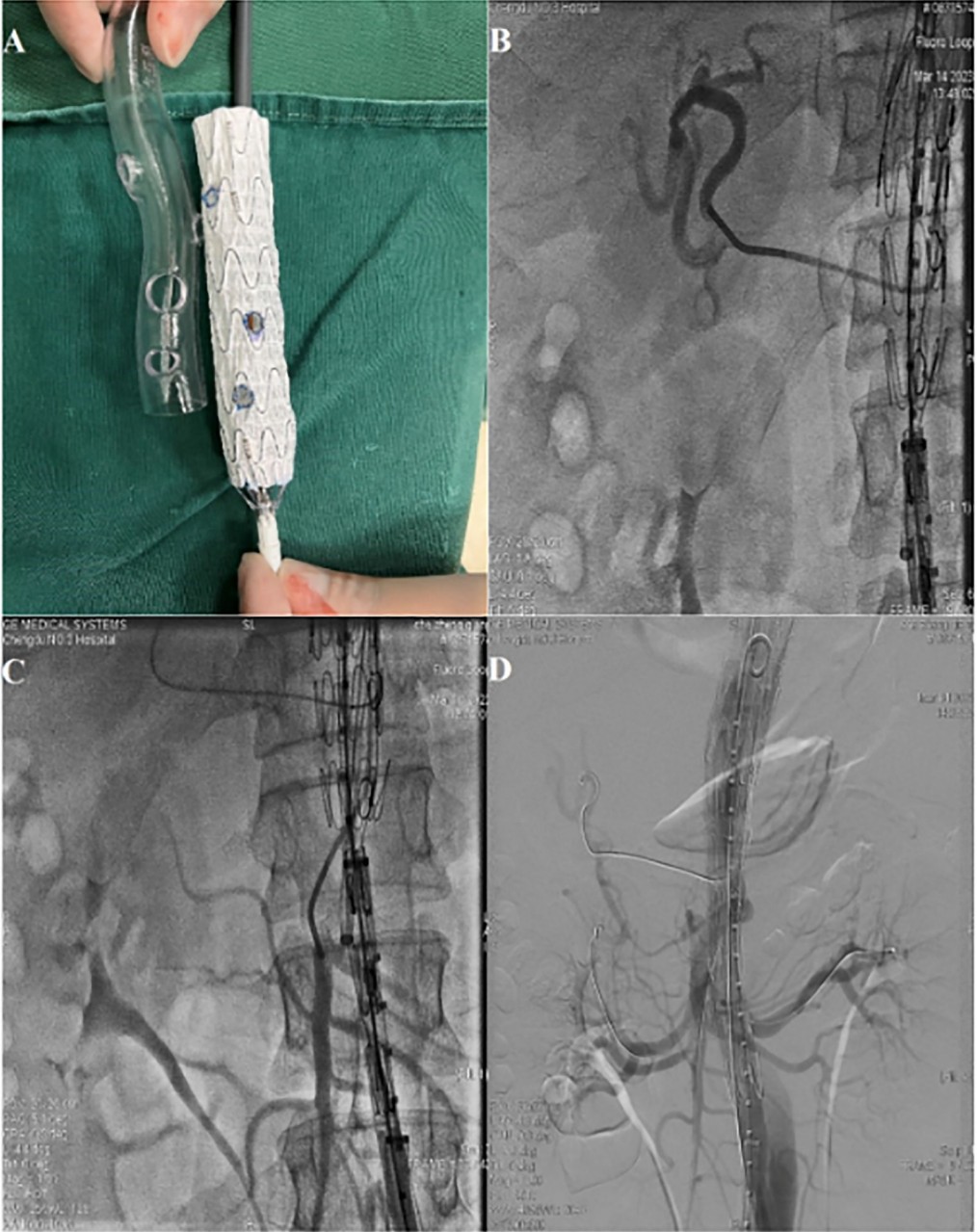

**Fig 1. Preparation of a four-vessel fenestrated endograft using a 3-dimensional (3D) printed model.** (A) The main aortic stent-graft was completely released in the 3D model, and the position of each fenestration was marked and a wire was sutured around its edge. (B) Super-selected celiac trunk. (C), Super-selective superior mesenteric artery. (D) Super-selective renal artery.

## Outcome measures

Outcome measures included early and late patient outcomes. Early outcome measures included procedural success and early complications. Technical success was defined as the successful reconstruction of the visceral vasculature of the abdominal aorta, isolation of distal tears, patency of target vessels and no any endoleak. Early complications were defined as

events that occurred within 30 days post-procedure and included mortality within 30 days post-procedure, stroke, spinal cord ischemia, organ failure, and myocardial infarction. Late outcomes were comprised of late aortic events, aortic remodelling, re-intervention, and late mortality. Late aortic events were defined as new stent-induced dissecting tears, target vessel occlusion, and endoleak. Endoleak was defined based on the definitions of the Society for Vascular Surgery [6]. Aortic remodelling was defined as true lumen (TL) re-expansion and false lumen (FL) retraction with concomitant thrombosis.

### Follow-up

Follow-up was performed 3 months, 6 months, and 12 months post-procedure, and every year thereafter. At each follow-up, each segment of the aorta was examined by CTA imaging. Target vessel occlusion, stent integrity, stent migration, the presence of endoleaks, and aortic remodelling were evaluated. Aortic remodeling was evaluated by FLT, and the maximum diameters of thoracic aorta and abdominal aorta were measured. The diameters of the FL and TL were recorded at the same segment. The FL was imaged and classified to have either no thrombosis, partial thrombosis, or complete thrombosis.

### Statistical methods

Continuous variables with normal distribution are expressed as mean ± standard deviations, while with non-normal distribution represented by median and interquartile range. Categorical variables are presented as frequency and percentage. The linear mixed model was used to explore the differences in the diameter of the aorta in at different time points. The overall survival and aortic specific survival curves were generated using the Kaplan–Meier method. Two-sided P value less than 0.05 means was statistically significant. All statistical analyses were performed using statistical software SPSS26.0 (IBM Inc., Armonk, NY).

## Results

### Baseline characteristics

As shown in Table 1, a total of 36 patients (21 males, median age 64.5, range 44.5–71.8) were enrolled in the present study according to the including criteria. All patients had a history of hypertension, eight patients (22.2%) had chronic obstructive pulmonary disease (COPD), three patients (8.8%) had a history of stroke, and three patients (8.8%) were diabetic. Two patients (5.6%) had a history of renal insufficiency. Hyperlipidaemia and myocardial infarction were observed in 1.6% of the patients. Patients determined to be suitable for F-EVAR had the following conditions: FL progression (66.7%), symptoms (13.9%), stent-induced new entry tears (11.1%), post-dissection aneurysm (11.1%), and organ ischemia (8.3%).

### Operative details

In all patients, composite fenestrated systems were applied. The procedure was carried out under general anaesthesia in all patients. The mean operation time was 6.5 ± 1.3 hours, which included 2.1 hours of stent-graft customization and 2.9 hours of endovascular surgery (Table 2). The mean contrast volume was 225 mL, and the mean radiation dose was 2885 mGy. In total 89 visceral vessels (72 renal arteries (RA), 11 superior mesenteric arteries (SMA) and 6 celiac arteries (CA)) were targeted, 141 with fenestrations and 3 with embolisms. The three (8.3%) CA were embolized because the entry tear of the dissection was very close.

**Table 1. Initial patient characteristics.**

| Variables | Value |
|---|---|
| No. of patients | 36 |
| Age, years | 64.5 (44.5–71.8) |
| Male/female | 24/12 |
| Comorbidities | |
| Hypertension (n, %) | 36, 100 |
| Diabetes (n, %) | 3, 8.3 |
| Stroke (n, %) | 3, 8.3 |
| Renal insufficiency (n, %) | 2, 5.6 |
| Hyperlipidemia (n, %) | 1, 1.6 |
| Miocardial infarction (n, %) | 1, 1.6 |
| COPD (n, %) | 8, 22.2 |
| Reasons of F-EVAR | |
| Organ ischemia (n, %) | 3, 8.3 |
| False lumen progress (n, %) | 24, 66.7 |
| New entry tears caused by stent (n, %) | 4, 11.1 |
| Post-dissection aneurysm > 5cm (n, %) | 4, 11.1 |
| Abdominal pain associated with dissection (n, %) | 5, 13.9 |
| Interval between TEVAR and F-EVAR (months) | 10 (7.8–15.5) |

COPD, chronic obstructive pulmonary disease; F-EVAR, fenestrated endovascular abdominal aortic repair; TEVAR, thoracic endovascular aortic repair.

## Early outcomes

During the early outcome stage, the technical success rate was 97.2% (Table 3). One case failed due to the FL being too wide to reconstruct the left renal artery. The left renal artery was reconstructed by a parallel stent technique through a tear in the left common iliac artery. There were

**Table 2. Operative details.**

| Variables | Value |
|---|---|
| Operation time, h | 6.5 ± 1.3 |
| Stent-graft customization time, h | 2.1 ± 0.4 |
| Endovascular component time, h | 2.9 ± 0.6 |
| Intraoperative blood loss, mL | 258.5 ± 112.4 |
| Transfusion volume, mL | 192.5 (0–1000) |
| Contrast volume, mL | 224.6 ± 45.2 |
| Radiation dose, mGy | 2885.5±487.6 |
| Number of fenestrations (n, %) | |
| 3 | 3, 8.3 |
| 4 | 33, 91.7 |
| Branch arteries reconstructed (n, %) | |
| Celiac axis | 33, 91,7 |
| Superior mesenteric artery | 36, 100 |
| Inferior mesenteric artery | 0, 0 |
| Renal artery | 36, 100 |
| Accessory renal artery | 0, 0 |

**Table 3. Early outcomes.**

| Variables | Value |
|---|---|
| Technical success (n, %) | 35, 97.2 |
| 30-Day mortality (n, %) | 0, 0 |
| Stroke (n, %) | 1, 1.6 |
| Spinal cord ischemia | |
| Transient paraparesis (n, %) | 2, 5.6 |
| Permanent paraparesis (n, %) | 0, 0 |
| Miocardial infarction (n, %) | 0, 0 |
| Organ failure (n, %) | 0, 0 |

no cases of 30-day mortality, myocardial infarction, and organ failure after the procedure. One patient suffered from a stroke with left hemiplegia. After pharmacological and rehabilitation treatments, this particular patient recovered. Two patients developed transient spinal cord ischemia and recovered after conservative treatments. No patient in this study experienced permanent postoperative paraplegia.

## Late outcomes

During the late outcome stage post-procedure, the median follow-up period was 23 months (Table 4). Ten patients (27.8%) suffered from endoleak during this period (seven patients developed a type II endoleak and three patients developed an iliac artery Ib endoleak). Seven patients (19.4%) underwent reintervention after the procedure, of which three patients suffering from iliac artery Ib endoleak underwent balloon dilatation, and four patients with type II endoleak underwent inferior mesenteric artery and (or) lumbar artery embolisation. No target vessel occlusion occurred in any of these cases. The Kaplan-Meier survival curves of the overall survival rate and aortic-specific survival rate of the patients are shown in Fig 2. The 3-year overall survival rate was 75%. During the follow-up, five patients died: two patients from retrograde type A dissection (RTAD) at 11 and 18 months post-procedure and three patients from heart failure, multiple organ failure, and septic shock.

**Table 4. Late outcomes.**

| Variables | Value |
|---|---|
| Follow-up (months) | 23 (11–32.8) |
| Late aortic events | |
| Endoleak | |
| Type I (n, %) | 3, 8.3 |
| Type II (n, %) | 7, 19.4 |
| Type III (n, %) | 0, 0 |
| Reintervention (n, %) | 7, 19.4 |
| Target vessel occlusion (n, %) | 0, 0 |
| Late death | |
| Aortic related (n, %) | 2, 5.6 |
| Aortic unrelated (n, %) | 3, 8.3 |

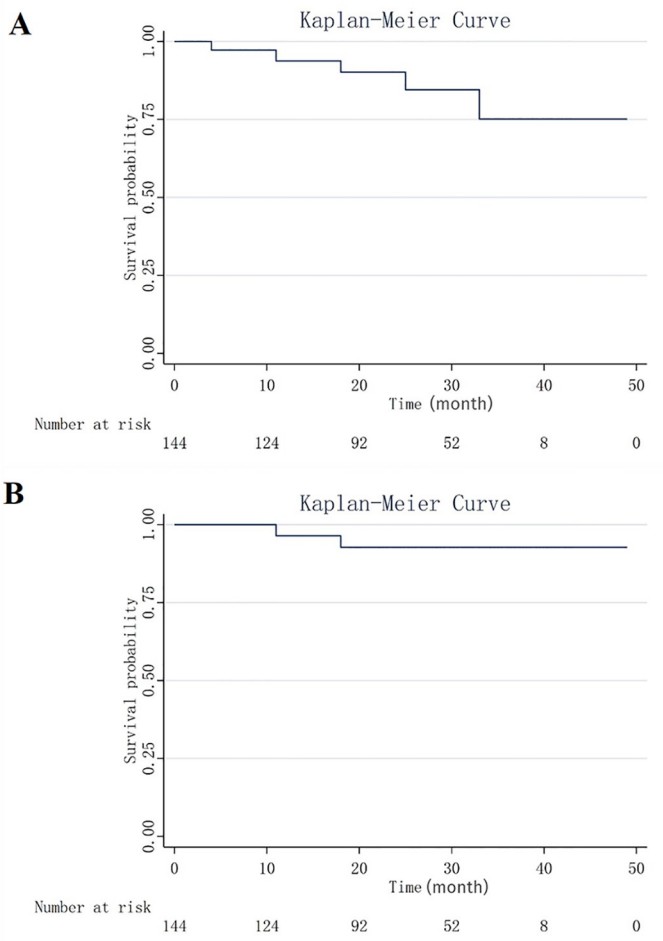

**Fig 2. Kaplan-Meier curves (Time units) showing overall survival (A) and aorta-specific survival (B).**

## Aortic remodeling

A total of 36 patients were evaluated for aortic remodelling at 3, 6, and 12 months post-procedure. This is the CTA image of a patient before and after the procedure (Fig 3). The changes to the aortic diameter pre- and post-procedure were shown in Table 5. CTA imaging of one year after operation indicated that all the patients had complete FLT in their thoracic and abdominal aortas, respectively. During the follow-up, the TL diameters at the thoracic aortic level (P < 0.021) and abdominal aortic level (P < 0.001) were significantly increased. The FL diameter at the same segmental level was also significantly smaller than it was pre-procedure (P < 0.001). The diameter of the aorta decreased significantly at both segmental levels (P < 0.001; P < 0.002).

## Discussion

The results of this study indicate that F-EVAR treatment of distal tears of chronic DeBakey IIIb aortic dissection had excellent early and late outcomes. The technical success rate in this study was 97.2%. In the early outcome stage, there was no 30-day mortality, organ failure, and permanent paraplegia. The rate of stroke was only 1.6%. During the postoperative follow-up,

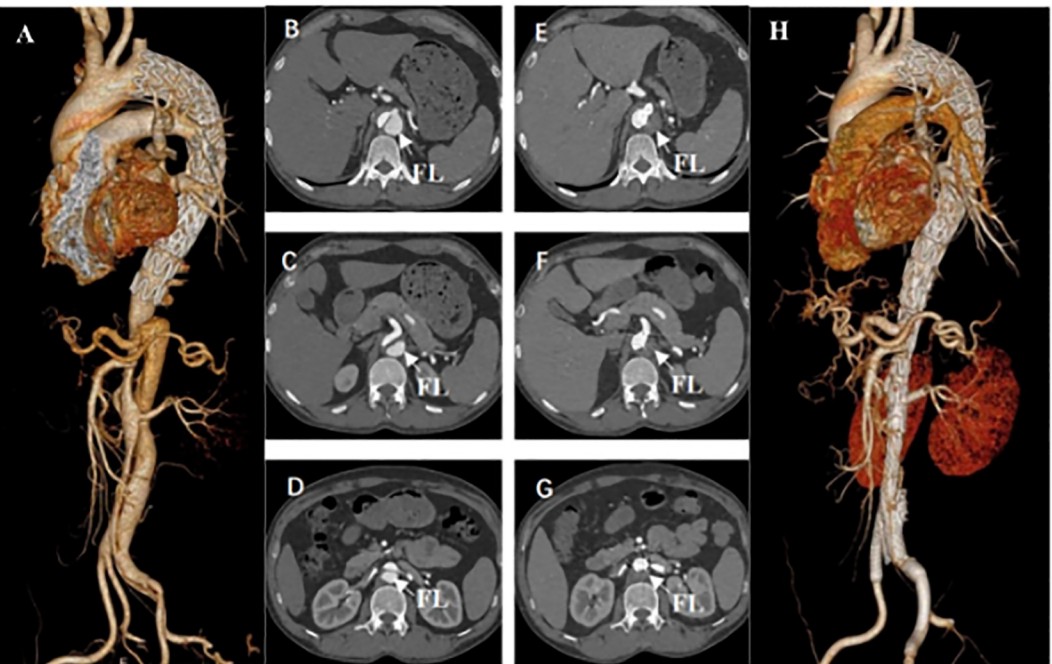

**Fig 3.** Computed tomography angiogram (CTA) of a patient before (A-D) and after (E-H) fenestrated endovascular abdominal aortic repair (F-EVAR) (arrow, false lumen (FL)). (A) Distal residual dissection after thoracic endovascular aortic repair (TEVAR). (B-D) Preoperative CTA showing the true lumen (TL) and FL at the level of the celiac trunk, superior mesenteric artery and renal artery. (E-G) Postoperative CTA showing the TL and FL at the level of the celiac trunk, superior mesenteric artery and renal artery. (H) Three-dimensional reconstruction of the CTA revealing good aortic remodeling with FL shrinkage and TL expansion.

the rate of aortic remodelling was improved, suggesting that F-EVAR had improved outcomes compared to other treatment methods described in the previous medical literature [4, 7]. For example, Oikonomou et al. treated post-dissection aneurysms using a complex one-stage procedure which increased the risk of spinal cord ischemia and had a longer procedure time. The

**Table 5. Aortic remodeling after F-EVAR.**

| | Follow-up | | | | *P value |
|---|---|---|---|---|---|
| | Baseline | 3 Months | 6 Months | 1 Year | |
| | mean (SD) | mean (SD) | mean (SD) | mean (SD) | |
| True lumen (mm) | | | | | |
| TA level | 27.86 (4.90) | 28.42 (4.73) | 28.66 (4.24) | 28.28 (4.69) | 0.021 |
| AA level | 15.72 (4.46) | 20.81 (3.00) | 21.76 (2.67) | 21.20 (4.07) | <0.001 |
| False lumen (mm) | | | | | |
| TA level | 14.53 (7.09) | 12.39 (7.07) | 11.14 (7.31) | 11.48 (7.91) | <0.001 |
| AA level | 18.61 (5.33) | 13.06 (5.95) | 11.38 (6.14) | 13.12 (7.92) | <0.001 |
| Whole lumen (mm) | | | | | |
| TA level | 42.39 (8.49) | 40.78 (8.75) | 39.79 (8.18) | 39.84 (9.13) | <0.001 |
| AA level | 34.33 (5.95) | 33.86 (6.81) | 32.93 (6.92) | 33.92 (7.41) | 0.002 |

F-EVAR, fenestrated endovascular abdominal aortic repair; TA, thoracic aorta; AA, abdominal aorta; SD, Standard deviation.

*P value calculated by mixed linear model.

results of this study demonstrate that a staged repair method with strict follow-up protocols could not only reduce the time the patient is under anaesthesia but also effectively reduce the risk of spinal cord ischemia.

At present, whether aortic dissection distal tears need to be treated or not depends on the patient's condition [8, 9]. If the distal tears are located above the abdominal trunk, there is no doubt that it can be repaired in one stage. If the distal tears of the visceral vascular area of abdominal aorta is involved, it is considered that the primary repair of all the tears is too radical and the surgical risk is too high [10, 11]. According to previous studies, strict follow-up management can effectively reduce the risk; if there are changes in the disease (rapid dilation of the aorta and organ ischemia) can be timely intervention [9]. As the results of this study indicate, a staged F-EVAR procedure could have advantages over single-stage procedures as it can reduce the risk of surgical complications to patients.

At present, several procedures are available to treat distal tears of aortic dissection, examples include The False Lumen Interventions to promote Remodeling and Thrombosis (FLIRT) concept [3], the "cork in the bottle neck" Technique [12–14], the Provisional ExTension To Induce Complete ATtachment (PETTICOAT) [15], FL using atrial septal occluder [16], The Candy-Plug Technique [17–19] and so on. For aortic remodelling, the use of covered stents to completely restore single-lumen blood flow has been proven to be a highly effective treatment method. Evidence of aortic remodelling is closely related to an improved long-term prognosis for patients and enhanced aortic remodelling has been reported to have an even better prognosis [3, 20]. Despite these advantages, the F-EVAR procedure has not been a popular option for distal tear repair in aortic dissection.

On the one hand, the operation is difficult, because the distal tears of the dissection frequently involve the visceral area of the abdominal aorta [9]. To repair tears in the visceral area of the aorta, it is necessary to reconstruct the visceral vasculature. At present, the preferred surgical method is custom-made branched or fenestrated stent grafts, but these stents are unavailable in China [21]. Therefore, visceral vasculature can only be reconstructed by a complex in-vitro pre-fenestration technique which requires advanced experience and skill of the performing surgeons. Recently, with the rise of 3D printing technology, a 3D printed aortic model can not only improve the accuracy of aortic fenestration methods but also reduce the complexity of the procedure. The results of this study correlate with previous studies and confirm that 3D aortic models improve patient outcomes as none of the target vessels in the visceral area of any surviving patient was observed to be occluded during follow-up [22–24].

On the other hand, the surgical risks, particularly the risk of paraplegia during F-EVAR procedures may also be a reason why the procedure is utilised less often. Once paraplegia occurs, the outcome is disastrous for patients and their families. Studies have shown an increased correlation between the risk of paraplegia and longer lengths of aortic stents [8, 25]. For patients who require a longer segment of the aorta to be covered in stents, the risk of paraplegia can be greatly reduced provided that meticulous surgical techniques are employed, including measures such as careful spinal cord protection, thorough preoperative evaluation, and the expertise of experienced surgeons specialising in complex aortic interventions. Oikonomou et al. (2014) reported that only four (12.6%) patients with dissecting aneurysms treated with F/Br-TEVAR had developed transient spinal cord ischemia and had no permanent paraplegia [7]. Kuzniar et al. reported similar results with three (11.1%) patients who had transient spinal cord ischemia and no permanent paraplegia [26]. The results of this study show that only two (5.6%) of 36 patients had transient spinal cord ischemia and no permanent paraplegia. This indicates that the methods of this study resulted in the best outcomes for paraplegia of the three studies examined. These results are due to the following reasons: procedures were staged rather than singular; during postoperative management, blood pressure was maintained

at a normal high value with a haemoglobin level at $> 10$ mg/dl; the left subclavian artery and internal iliac artery were retained.

In addition, there is a high risk of re-intervention associated with F-EVAR involving the aorta. As previously reported, the incidence of F/B-TEVAR re-intervention after aortic dissection was 22.5% to 52% [26]. The rate of re-intervention for this study was 19.4%. Interestingly, most of the re-intervention events patients experienced were due to type I and type II endoleak. In most cases, type II endoleaks do not require treatment, whereas type I endoleaks typically only necessitate balloon dilatation treatments under local anaesthesia. This procedure carries a low risk of complications to the patient. Considering the long-term benefits and patient outcomes, performing balloon dilation is regarded as a reasonable approach in the majority of cases.

Staged surgery has been reported to reduce the risk of paraplegia, which may be due to the establishment of collateral circulation, but optimal intervals between the two stages of staged surgery are still unclear [27]. In the patient cohort of this study, no patient developed permanent paraplegia and the interval between the first TEVAR and the second F-EVAR was an average of 10 months (7.8–15.5 months). This indicated that a 10-month interval between the two procedures was sufficient for the formation of spinal collateral circulation. Nevertheless, further validation of this approach requires a larger prospective dataset.

In the examined patient cohort, there were two aortic-related deaths (5.6%), both due to retrograde type A dissection (RTAD). This incidence is higher than a recently published meta-analysis (3%) [28]. he two patients who developed the RTAD had the dissection located close to the opening of the left subclavian artery ($< 1$cm); as a result, the left subclavian artery was reconstructed during the first TEVAR procedure and the proximal landing zone of stent graft was located in area 1–2. According to the literature, the proximal placement of stent grafts in the 0–2 zone of the aorta is an important risk factor for RTAD [29]. Another potential risk factor may be the guide wire and (or) sheath operation of the arch [30]. For these two patients, TEVAR was performed in 2016 in the absence of an integrated stent and a needle was used during the *in-situ* fenestration technique to reconstruct the left subclavian artery. This resulted in the guide wire and (or) sheath procedure being more complex than the standard TEVAR, possibly leading to aortic wall damage. Tsilimparis et al. compared the results of fenestrated and branched stents in twenty-nine patients and found that the use of fenestrated stents had a higher mortality rate compared to branched stents (20% vs 0%) [31]. In this study, patients who had reconstruction to the left subclavian artery after the introduction of integrated stents did not develop RTAD.

Several limitations are associated with the results obtained from this cohort of patients. First, this was a single-centre retrospective study. The reliability of the results could be improved by incorporating data from multiple institutions. In addition, this study evaluated aortic remodelling by measuring the aortic maximum diameter and FLT. Relying solely on these methods could potentially result in numerous variations or inconsistencies. Including techniques to measure volume analysis or the direct assessment of aortic hemodynamic effects would further evaluate the extent of patient aortic remodelling.

## Conclusion

In conclusion, the use of 3D-printed model-guided physician-modified fenestrated endografts to treat distal tears of chronic DeBakey IIIb aortic dissection was shown to be an overall safe and effective treatment for improved patient outcomes. As a result, it is suggested that distal tears of DeBakey type IIIb aortic dissection should be followed up with strict treatment and observation methods after the treatment of an aortic proximal tear.

## Supporting information

**S1 Checklist.** *PLOS ONE* **clinical studies checklist.**
(DOCX)

**S2 Checklist. STROBE statement—Checklist of items that should be included in reports of observational studies.**
(DOCX)

## Author Contributions

**Software:** Wei Liu.

**Writing – original draft:** Chi Cui, Wei Liu.

**Writing – review & editing:** Bisi Wang.

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
