## [Decision Letter · Decision Letter 0]

29 Aug 2023

PONE-D-23-18227Outcomes of Fenestrated Endovascular Abdominal Aortic Repair in Distal Entry Tears of Chronic DeBakey IIIb Aortic DissectionPLOS ONE

Dear Dr. Liu,

Thank you for submitting your manuscript to PLOS ONE. After careful consideration, we feel that it has merit but does not fully meet PLOS ONE’s publication criteria as it currently stands. Therefore, we invite you to submit a revised version of the manuscript that addresses the points raised during the review process.

We look forward to receiving your revised manuscript.

Kind regards,

Alessandro Leone, MD

Academic Editor

PLOS ONE

6. We note that Figures 1 and 2 in your submission contain copyrighted images. All PLOS content is published under the Creative Commons Attribution License (CC BY 4.0), which means that the manuscript, images, and Supporting Information files will be freely available online, and any third party is permitted to access, download, copy, distribute, and use these materials in any way, even commercially, with proper attribution. For more information, see our copyright guidelines: http://journals.plos.org/plosone/s/licenses-and-copyright.

a. You may seek permission from the original copyright holder of Figures 1 and 2 to publish the content specifically under the CC BY 4.0 license.

Reviewers' comments:

Reviewer's Responses to Questions

**Comments to the Author**

1. Is the manuscript technically sound, and do the data support the conclusions?

Reviewer #1: Yes

Reviewer #2: Partly

2. Has the statistical analysis been performed appropriately and rigorously? 

Reviewer #1: Yes

Reviewer #2: Yes

3. Have the authors made all data underlying the findings in their manuscript fully available?

Reviewer #1: Yes

Reviewer #2: Yes

4. Is the manuscript presented in an intelligible fashion and written in standard English?

Reviewer #1: Yes

Reviewer #2: No

5. Review Comments to the Author

Reviewer #1: Dear the authors of the manuscript entitled "Outcomes of Fenestrated Endovascular Abdominal Aortic Repair in Distal Entry Tears of Chronic DeBakey IIIb Aortic Dissection"

Thank you for writing this manuscript which I believe that it highlights an important issue which we need further evidence in the best pathway to utilize in the management of this critical subset of patients

I believe the method of utilizing 3-dimensional (3D) printed aortic model-guided fenestrated stent in the treatment of distal tears of chronic DeBakey IIIb aortic dissection after thoracic endovascular aortic repair (TEVAR) is a valuable technique to consider

I have couple of comments to mention here:

1. I understand the indications you presented for the utilization of this technique, but could you elaborate more about the functional status of the abdominal organs before and after treatment (Lactate, Liver function, Kidney function)

2. Can you elaborate if there is any place for this modality to be used in the acute setting in association with TEVAR?

3. was there any difference between the patient cohort who had aortic remodeling after this technique in terms of secondary prevention treatment such as taking b blockers?

Thank you

Reviewer #2: This is a retrospective study in which the authors evaluated the clinical outcomes of a 3-dimensional (3D) printed aortic model-guided fenestrated stent in the treatment of distal tears of chronic DeBakey IIIb aortic dissection after thoracic endovascular aortic repair (TEVAR). The study population included 36 patients with chronic DeBakey IIIb aortic dissection who underwent TEVAR and fenestrated endovascular abdominal aortic repair (F-EVAR) from April 2014 to December 2022.

The early outcomes evaluated were procedural success (successful reconstruction of the visceral vasculature of the abdominal aorta, isolation of distal tears, patency of target vessels and no any endoleak) and early complications (mortality within 30 days post-procedure, stroke, spinal cord ischemia, organ failure, and myocardial infarction). Late outcomes included late aortic events (new stent-induced dissecting tears, target vessel occlusion, and endoleak), aortic remodelling (true lumen re-expansion and false lumen retraction with concomitant thrombosis), re-intervention and late mortality. The technical success rate was 97.2% and there were no cases of 30‐ day mortality, myocardial infarction, permanent paraparesis, or organ failure. At follow-up, surviving patients showed significant false and true lumen remodelling. The authors concluded that F-EVAR was a safe and feasible operation to repair all distal tears of chronic DeBakey IIIb aortic dissection.

The topic of this study is very interesting.

However, there are some points of discussion:

1. The manuscript needs an English revision.

2. The number of patients included in the study should be explain in the “Method section”, both in the abstract and in the text.

3. In the section “Procedures”, the authors explain the technical procedure of construction of the prosthesis and the intraoperative procedure of implantation. In order to be clearer in the description of the different procedures in the text, I suggest to split the section in two subsections.

4. In the section “Outcome measures”, the sentence “Late outcomes were comprised of late aortic events, aortic remodelling, re-intervention, and late mortality” was repeated.

5. The “Table I” showed the type of procedures performed for each patient at the level of each abdominal vessel (celiac artery, superior mesenteric artery, left renal artery and right renal artery). The title of this table is “Dissection characteristics”. I suggest to choose a title that better reflects the contents of the table and to change the table style to be clearer to the reader. Moreover, the table should be more extensively explained in the text.

6. In the “Table II”, the variable “Reasons of F-EVAR” should be replaced by “Indication to F-EVAR”.

7. The style of the Tables II and III should be changed, especially reporting more clearly what the values refer.

8. In the text the reference to “Figure 2”, which shows the CT images of a patient before and after the procedure, should not be referred in the section "Procedures", after the explanation of the interventional procedure. This reference should be moved to “Aortic remodelling” section.

9. The “Figure 3” show the Kaplan-Meier curves of overall survival and aorta-specific survival. Detailed data on survival with patients at risk should be added.

10. The bibliography includes some dated articles. I suggest to refer to more recent articles.

11. In the section "Discussion", the authors should comment more extensively the comparison with other techniques available for the treatment of this pathology, including traditional surgery.

6. PLOS authors have the option to publish the peer review history of their article (what does this mean?). If published, this will include your full peer review and any attached files.

Reviewer #1: **Yes: **Salah Eldien Altarabsheh

Reviewer #2: No

---

## [Author Response · Author response to Decision Letter 0]

8 Oct 2023

Dear Editors and Reviewers, 

Thank you for your decision and constructive comments on my manuscript. We have carefully considered the suggestion of Reviewer and make some changes. We have tried our best to improve and made some changes in the manuscript. The yellow part that has been revised according to your comments. Revision notes, point-to-point, are given as follows: 

1. The review’s comment: I understand the indications you presented for the utilization of this technique, but could you elaborate more about the functional status of the abdominal organs before and after treatment (Lactate, Liver function, Kidney function)

The authors’ answer: According to the results of blood examination before and after operation, this operation (fenestrated endovascular abdominal aortic repair, F-EVAR) does not affect the function of abdominal organs. If patients are operated on because of abdominal organ ischemia, the abdominal organ function can be improved after operation.

2. The review’s comment: Can you elaborate if there is any place for this modality to be used in the acute setting in association with TEVAR?

The authors’ answer: F-EVAR can be used for emergency treatment as long as patient can wait 3-4 hours. Because we need to wait for 3-dimensional (3D) printed model of the aorta, which is usually 3-4 hours. Compared with commercial custom stents that need to wait 1-2 months, we think 3D printed aortic model-guided fenestrated stent is more suitable for emergency treatment. 

3. The review’s comment: Was there any difference between the patient cohort who had aortic remodeling after this technique in terms of secondary prevention treatment such as taking b blockers?

The authors’ answer: In all the dissecting patients in our center, we routinely take b blockers orally, because there was a lot of evidence that long-term oral b blockers could improve the prognosis of dissecting patients. Therefore, it is not clear whether the absence of oral b blockers after F-EVAR will affect aortic remodeling.

4. The review’s comment: The manuscript needs an English revision.

The authors’ answer: We apologize for the poor language of our manuscript. We worked on the manuscript for a long time and the repeated addition and removal of sentences and sections obviously led to poor readability. We have now worked on both language and readability and have also involved native English speakers for language corrections. We really hope that the flow and language level have been substantially improved.

5. The review’s comment: The number of patients included in the study should be explain in the “Method section”, both in the abstract and in the text.

The authors’ answer: The number of patients included in the study has been described in the “Method section”, both in the abstract and in the text.

6. The review’s comment: In the section “Procedures”, the authors explain the technical procedure of construction of the prosthesis and the intraoperative procedure of implantation. In order to be clearer in the description of the different procedures in the text, I suggest to split the section in two subsections.

The authors’ answer: We have divided the section “Procedures” into two parts. The first part was the construction of fenestrated stents, and the second part was the implantation process in vivo.

7. The review’s comment: In the section “Outcome measures”, the sentence “Late outcomes were comprised of late aortic events, aortic remodelling, re-intervention, and late mortality” was repeated.

The authors’ answer: We are very grateful to the reviewers for their careful review and the repetitive sentences have been deleted.

8. The review’s comment: The “Table I” showed the type of procedures performed for each patient at the level of each abdominal vessel (celiac artery, superior mesenteric artery, left renal artery and right renal artery). The title of this table is “Dissection characteristics”. I suggest to choose a title that better reflects the contents of the table and to change the table style to be clearer to the reader. Moreover, the table should be more extensively explained in the text.

The authors’ answer: We have refined the title "Table I" and explained it in detail in the results section. The serial number of the Table is also adjusted. 

9. The review’s comment: In the “Table II”, the variable “Reasons of F-EVAR” should be replaced by “Indication to F-EVAR”.

The authors’ answer: We have made corrections. 

10. The review’s comment: The style of the Tables II and III should be changed, especially reporting more clearly what the values refer.

The authors’ answer: We have adjusted the order of the original Tables Ⅱ and Ⅲ. Separate the early outcome from the late outcome, so that the reader has a better understanding of what each variable represents.

11. The review’s comment: In the text the reference to “Figure 2”, which shows the CT images of a patient before and after the procedure, should not be referred in the section "Procedures", after the explanation of the interventional procedure. This reference should be moved to “Aortic remodelling” section.

The authors’ answer: We have put “Figure 2” into the to “Aortic remodelling” section. 

12. The review’s comment: The “Figure 3” show the Kaplan-Meier curves of overall survival and aorta-specific survival. Detailed data on survival with patients at risk should be added.

The authors’ answer: We have added detailed data on survival with patients at risk.

13. The review’s comment: The bibliography includes some dated articles. I suggest to refer to more recent articles.

The authors’ answer: We have replaced those dated literatures with recent articles.

14. The review’s comment: In the section "Discussion", the authors should comment more extensively the comparison with other techniques available for the treatment of this pathology, including traditional surgery.

The authors’ answer: Other methods for the treatment of residual aortic dissection have been introduced in detail in the third paragraph of the discussion section. Traditional surgical procedures are not discussed in this article because this cohort was not suitable for patients undergoing traditional open surgery. It has been explained in the method section. 

We tried our best to improve the manuscript and made some changes in the manuscript. 

We appreciate for Editors/Reviewers’ warmwork earnestly, and hope that the correction will meet with approval. Once again thank you very much for your comments and suggestiongs.

Yours sincerely, 

Wei Liu

---

## [Decision Letter · Decision Letter 1]

8 Jan 2024

PONE-D-23-18227R1Outcomes of Fenestrated Endovascular Abdominal Aortic Repair in Distal Entry Tears of Chronic DeBakey IIIb Aortic DissectionPLOS ONE

Dear Dr. Liu,

Thank you for submitting your manuscript to PLOS ONE. After careful consideration, we feel that it has merit but does not fully meet PLOS ONE’s publication criteria as it currently stands. Therefore, we invite you to submit a revised version of the manuscript that addresses the points raised during the review process.

We look forward to receiving your revised manuscript.

Kind regards,

Alessandro Leone, MD

Academic Editor

PLOS ONE

Journal Requirements:

Reviewers' comments:

Reviewer's Responses to Questions

**Comments to the Author**

1. If the authors have adequately addressed your comments raised in a previous round of review and you feel that this manuscript is now acceptable for publication, you may indicate that here to bypass the “Comments to the Author” section, enter your conflict of interest statement in the “Confidential to Editor” section, and submit your "Accept" recommendation.

Reviewer #1: All comments have been addressed

Reviewer #3: All comments have been addressed

Reviewer #4: (No Response)

2. Is the manuscript technically sound, and do the data support the conclusions?

Reviewer #1: Yes

Reviewer #3: Yes

Reviewer #4: Yes

3. Has the statistical analysis been performed appropriately and rigorously? 

Reviewer #1: Yes

Reviewer #3: Yes

Reviewer #4: Yes

4. Have the authors made all data underlying the findings in their manuscript fully available?

Reviewer #1: Yes

Reviewer #3: No

Reviewer #4: Yes

5. Is the manuscript presented in an intelligible fashion and written in standard English?

Reviewer #1: Yes

Reviewer #3: Yes

Reviewer #4: Yes

6. Review Comments to the Author

Reviewer #1: Dear the authors of the manuscript entitled "Outcomes of Fenestrated Endovascular

Abdominal Aortic Repair in Distal Entry Tears of Chronic DeBakey IIIb Aortic Dissection"

Thank you for taking consideration for all the reviews comments

I have no concerns about this manuscript

Thank you

Reviewer #3: Minor proof-readig for some small errors in the text. Please do indicate tipe uints in your KM curves. Table one incorporates table 2, please do corrct the styling. Make available de-identified data. Table 2 endolieak type? I will rather personally prefer a line graph instead of table 3.

Reviewer #4: I read the reviewed version of the article “Outcomes of fenestrated endovascular abdominal aortic repair in distal entry tears of chronic DeBakey IIIb aortic dissection”, which discusses the outcome of patients with chronic DeBakey IIIb aortic dissection who have already undergone thoracic endovascular aortic repair (TEVAR) with a residual distal entry tears treated with a 3-dimensional (3D) printed model-guided physician-modified stent graft (PMSG) for fenestrated endovascular abdominal aortic repair (FEVAR).

The study is very interesting because currently in the literature there are few data on the treatment of distal entry tears of chronic DeBakey IIIb aortic dissection with this type of grafts.

Despite the comments raised in the previous review have been adequately addressed, I think that this scientific article requires further improvement before it can be published:

- In the “Procedures” section, would it be possible to provide more details on the creation of the 3D aortic template?

- In the “Baseline characteristics” section, is it possible to specify the patients’ symptoms?

- In Table 1 “Initial patient characteristics”, and particularly in the comorbidities section, is it possible to specify if there were patients with connective tissue disease?

- Details regarding the procedures should be added in the “Operative details” section (for example the time needed to prepare the grafts and the duration of the entire procedures, how much contrast volume was used during the procedures and how many transfusions were performed in case of blood loss, and other information that may be useful…)

- Table 2 “Dissection characteristics with details of reconstruction of brunch vessels” should be completely modified as it is not very intelligible in this format

In addition to this revision, it may be useful to add some small technical advice in your article in case one of these situations has happened to you:

- Have you ever had technical errors that led you to change your surgical strategy over the years?

- Do you have any special precautions in case of severe aortic tortuosity?

7. PLOS authors have the option to publish the peer review history of their article (what does this mean?). If published, this will include your full peer review and any attached files.

Reviewer #1: **Yes: **Salah Eldien Altarabsheh

Reviewer #3: **Yes: **Rafik Margaryan

Reviewer #4: No

---

## [Author Response · Author response to Decision Letter 1]

10 Jan 2024

Dear Editors and Reviewers, 

Thank you for your decision and constructive comments on my manuscript. We have carefully considered the suggestion of Reviewer and make some changes. We have tried our best to improve and made some changes in the manuscript. The yellow part that has been revised according to your comments. Revision notes, point-to-point, are given as follows: 

1. The review’s comment: Minor proof-readig for some small errors in the text. 

The authors’ answer: We are very grateful to the reviewers’ reminders and have corrected the errors in the manuscript.

2. The review’s comment: Please do indicate tipe (time?) uints in your KM curves.

The authors’ answer: We have made corrections. 

3. The review’s comment: Table one incorporates table 2, please do corrct (correct?) the styling.

The authors’ answer: Thank you very much for reminding us that we have made adjustments.

4. The review’s comment: Table 2 (4?) endolieak (endoleak?) type?

The authors’ answer: We have added endoleak type in Table 4.

5. The review’s comment: In the “Procedures” section, would it be possible to provide more details on the creation of the 3D aortic template?

The authors’ answer: We have added our detailed steps to make the 3D aortic template in the “Procedures” section.

6. The review’s comment: In the “Baseline characteristics” section, is it possible to specify the patients’ symptoms?

The authors’ answer: We have made corrections.

7. The review’s comment: In Table 1 “Initial patient characteristics”, and particularly in the comorbidities section, is it possible to specify if there were patients with connective tissue disease?

The authors’ answer: These patients were not associated with connective tissue disease.

8. The review’s comment: Details regarding the procedures should be added in the “Operative details” section (for example the time needed to prepare the grafts and the duration of the entire procedures, how much contrast volume was used during the procedures and how many transfusions were performed in case of blood loss, and other information that may be useful…) Table 2 “Dissection characteristics with details of reconstruction of brunch vessels” should be completely modified as it is not very intelligible in this format

The authors’ answer: We have readjusted and simplified the " Table 2" and added a lot of surgery-related information.

9. The review’s comment: Have you ever had technical errors that led you to change your surgical strategy over the years?

The authors’ answer: We once encountered a case of left renal artery catheterization failure, which was rescued by parallel stent technique. It is described in detail in “ early outcomes ” section.

10. The review’s comment: Do you have any special precautions in case of severe aortic tortuosity?

The authors’ answer: We usually evaluate whether the distortion of the aorta can be operated on by endovascular surgery before operation. If it is not suitable for general open surgery, but so far, our center has not encountered this kind of situation.

We tried our best to improve the manuscript and made some changes in the manuscript. 

We appreciate for Editors/Reviewers’ warmwork earnestly, and hope that the correction will meet with approval. Once again thank you very much for your comments and suggestiongs.

---

## [Editor Report · Decision Letter 2]

25 Jan 2024

Outcomes of Fenestrated Endovascular Abdominal Aortic Repair in Distal Entry Tears of Chronic DeBakey IIIb Aortic Dissection

PONE-D-23-18227R2

Dear Dr. Liu

We’re pleased to inform you that your manuscript has been judged scientifically suitable for publication and will be formally accepted for publication once it meets all outstanding technical requirements.

Kind regards,

Alessandro Leone, MD

Academic Editor

PLOS ONE
---

## [Editor Report · Acceptance letter]

17 Feb 2024

PONE-D-23-18227R2 

PLOS ONE

Dear Dr. Liu, 

I'm pleased to inform you that your manuscript has been deemed suitable for publication in PLOS ONE. Congratulations! Your manuscript is now being handed over to our production team.

Kind regards, 

on behalf of

Dr. Alessandro Leone 

Academic Editor

PLOS ONE